# Attachment Styles as Protective and Amplifying Factors in Caregiver Psychological Distress: A Multicenter Cross-Sectional Study in Oncology and Chronic Disease Caregiving

**DOI:** 10.3390/healthcare13202612

**Published:** 2025-10-17

**Authors:** Ipek Özönder Ünal, Miray Pirincci Aytac, Nur Özgedik Turhan, Yunus Taylan, Çağlar Ünal, Atakan Topcu, Eyyüp Cavdar, Erkan Özcan, Tomris Duymaz, Tonguc Demir Berkol

**Affiliations:** 1Department of Psychiatry, Kartal Dr. Lutfi Kirdar City Hospital, 34865 Istanbul, Turkey; 2Department of Psychiatry, Sancaktepe Şehit Prof. Dr. İlhan Varank Training and Research Hospital, 34785 Istanbul, Turkey; mirayaytac23@gmail.com; 3Department of Psychiatry, Bolu Abant İzzet Baysal Training and Research Hospital for Psychiatry, 14030 Bolu, Turkey; drnurozgedik@gmail.com; 4Department of Psychiatry, Sultanbeyli State Hospital, 34935 Istanbul, Turkey; yunustaylan99@gmail.com; 5Division of Medical Oncology, Department of Internal Medicine, Haydarpaşa Numune Research and Training Hospital, 34668 Istanbul, Turkey; caglarunal5@gmail.com (Ç.Ü.); atakantopcu@hotmail.com (A.T.); 6Division of Medical Oncology, Department of Internal Medicine, Adıyaman Research and Training Hospital, 02000 Adıyaman, Turkey; eyyupcavdar@hotmail.com; 7Division of Medical Oncology, Department of Internal Medicine, Kastamonu Research and Training Hospital, 37150 Kastamonu, Turkey; theerkanozcan_57@hotmail.com; 8Department of Physiotherapy & Rehabilitation, Istanbul Bilgi University, 34387 Istanbul, Turkey; tomrisduymaz@gmail.com; 9Department of Psychiatry, Bakirkoy Research and Training Hospital for Psychiatry, 34147 Istanbul, Turkey; tberkol@gmail.com

**Keywords:** anxiety, attachment, caregiver burden, cancer, depression

## Abstract

**Background/Objectives:** Caregiving for patients with advanced cancer or chronic illness imposes substantial psychological burden, yet the role of caregiver attachment style in moderating this distress is underexplored. This multicenter, cross-sectional study investigates how attachment styles influence the relationship between psychological distress and caregiver burden in two populations: family caregivers of palliative-stage cancer patients and those supporting patients with chronic diseases. **Methods:** Across Turkey, 819 caregivers (412 cancer, 407 chronic disease) completed the Zarit Caregiver Burden Scale, DASS-21, and Relationship Scale Questionnaire. Hayes’ PROCESS macro was used to test the moderating role of attachment styles. **Results:** Cancer caregivers reported higher caregiver burden (Cohen’s d = 0.35, 95% CI [0.21, 0.49]) and stress than chronic disease caregivers, but lower depression. Secure attachment was negatively associated with burden, while preoccupied attachment was positively associated. For cancer caregivers, secure attachment buffered the impact of depression on burden (interaction B = –0.611, 95% CI [–0.861, –0.361]), whereas preoccupied attachment amplified it in both groups (cancer caregivers: B = 0.292, 95% CI [0.064, 0.520]; chronic disease caregivers B = 0.505, 95% CI [0.174, 0.836]). The final regression models explained 43.1% of variance in burden for the cancer group and 10.9% for the chronic disease group. **Conclusions:** Attachment styles are significant moderators of the relationship between psychological distress and caregiver burden. Secure attachment is a protective factor, while preoccupied attachment is a vulnerability factor. These findings underscore the need for attachment-informed psychosocial interventions tailored to specific caregiver profiles to mitigate distress.

## 1. Introduction

Cancer remains a formidable adversary in the global health landscape, affecting millions of individuals and their families worldwide [1]. Palliative care, an essential component in cancer management, aims to improve the quality of life for patients and their families, especially in advanced stages of the disease [2]. This caregiving role, frequently unchosen, can lead to numerous negative repercussions for family members, including physical and mental health issues, strained relationships, and reduced personal and professional opportunities [3,4,5].

This intricate tapestry of caregiving extends beyond cancer to myriad chronic conditions, such as dementia, stroke, and cardiovascular diseases, presenting a similar array of challenges [6,7,8,9]. Caregivers in these contexts are often confronted with significant physical demands and profound emotional and social upheavals, including sleep disturbances, anxiety, depression, and social isolation [10,11,12,13,14].

Caregiving is a foundational pillar of healthcare, yet the role imposes substantial burdens. Caregivers often experience physical ailments, such as sleep disturbances and cardiovascular issues, stemming from the constant demands of their responsibilities [10,11,12]. These physical challenges are compounded by significant emotional and social distress, including depression, anxiety, and social isolation [13,14].

The concept of caregiver burden is a multifaceted construct, shaped by the caregiver-patient relationship and the specific caregiving context [15,16,17,18]. To better understand this burden, it is crucial to examine the underlying relational dynamics. Among these, attachment styles—patterns rooted in perceptions of self and others—emerge as a key factor [19,20,21,22]. These styles are pivotal in shaping caregiver-patient interactions and can therefore substantially influence a caregiver’s psychological well-being.

Secure attachment, characterized by a positive view of oneself and others, tends to foster healthy, trust-rich relationships where individuals find comfort and willingness to forge close bonds, generally leading to psychological well-being and a lower propensity towards psychological disorders. Conversely, preoccupied attachment, defined by negative self-view and a positive view of others, engenders a hypersensitive alertness towards others’ proximity, leading to a clingy and dependent demeanor in relationships due to low self-worth, and may escalate vulnerability to certain psychopathologies. Dismissive-avoidant attachment manifests from a positive self-perception but a negative outlook towards others, often resulting in a cold and unexpressive demeanor in relationships. Lastly, fearful-avoidant attachment, arising from a negative perception of both oneself and others, fosters an avoidance of close relationships due to the fear of rejection, and is associated with a high incidence of psychopathology and interpersonal distress. Understanding these attachment styles could unveil critical insights into the complex dynamics between caregivers and patients, aiding in the development of more nuanced and effective support mechanisms [22].

The psychosocial burden of caregiving is deeply connected to the caregiver’s attachment style. Research consistently links insecure attachment styles (anxious and avoidant) to higher psychological distress, whereas secure attachment is associated with better emotional management and caregiving efficacy [23,24]. Further studies have explored how attachment influences caregiver motivations and experiences of burden, even in relation to physical proximity [20,25]. This body of evidence highlights the importance of integrating attachment theory into caregiving research and practice to develop more targeted and effective support mechanisms.

Attachment styles, rooted in perceptions of self and others, are fundamental to understanding the psychosocial dynamics of caregiving, particularly in the contexts of cancer and chronic disease. The primary styles—secure, preoccupied, avoidant, and fearful—are associated with distinct relational patterns that shape caregiving experiences and outcomes. A growing body of research confirms the profound impact of these styles, highlighting the necessity of applying this framework in both research and clinical practice [23,24,25]. Doing so allows for a clearer delineation of how attachment influences the caregiver’s role amidst the complexities of severe illness.

Research also highlights the complex interplay between emotional attachment and physical proximity in shaping caregiver burden. The work of Bei et al., for example, demonstrates how attachment orientations (such as anxiety and avoidance) interact with the physical distance from the care recipient to influence the caregiver’s experience [20]. This underscores the importance of considering both psychological and logistical factors, as they are often entangled in determining the overall burden of care.

In addressing critical gaps in the literature on the interplay between attachment styles and caregiver psychological wellness, our study methodically explores the pathways linking attachment styles with depression, stress, and anxiety, and their cumulative impact on caregiver burden. This focus is particularly pertinent among caregivers supporting patients with cancer and chronic diseases. Each attachment style, with its distinct challenges and relational dynamics, not only shapes the caregiving experience but also influences the psychological trajectory of the caregiver, adding complexity to an already demanding role. This research aims not merely to explore these relationships but to delineate how variations in attachment styles and their associated psychological challenges shape the caregiving landscape, particularly in terms of mental and emotional well-being. By constructing a detailed framework, we seek to enhance understanding of these dynamics and inform the development of caregiver support mechanisms that are precisely tailored to the unique challenges posed by different attachment styles. Beyond theoretical exploration, this study aspires to translate these insights into actionable, nuanced, and impactful interventions, ensuring caregivers are comprehensively understood, effectively supported, and empowered to navigate their multifaceted caregiving journeys with resilience and enhanced psychological well-being.

## 2. Materials and Methods

### 2.1. Study Design and Participants

This multicenter cross-sectional study included primary caregivers aged 18 to 65 years who had been providing care to patients with either cancer or chronic diseases for a minimum of six months. To ensure a more homogeneous participant group among cancer caregivers, inclusion criteria specified caregivers of patients with stage 4 solid tumors currently under palliative care and no longer receiving chemotherapy. These participants were recruited from oncology clinics at the point where patients were being followed by their oncologists and had been recommended for a transition to palliative care. For the chronic disease caregiver cohort, participants were actively involved in home-based care scenarios. These caregivers were supporting patients registered with the Tuzla State Hospital’s home care clinic, which provided regular home visits and treatment management from a hospital healthcare team. Chronic diseases in this study were defined as conditions persisting for more than three months, requiring ongoing medical attention and/or significantly limiting activities of daily living. These conditions, while potentially stabilized or managed, often impose long-term impacts on patients’ health and functioning.

Caregivers in the chronic disease cohort supported individuals with conditions such as Alzheimer’s disease (characterized by progressive memory loss and cognitive decline), stroke (often resulting in varying levels of physical, cognitive, and emotional impairment), intellectual disabilities (requiring assistance with learning, task management, and structured environments), and other disabilities with diverse care needs and challenges. These conditions were selected due to their inherent demand for substantial, ongoing caregiving, which profoundly impacts caregivers’ physical and psychological well-being, offering a robust context for examining caregiver burden and related psychosocial dynamics. The caregiving role in this study was defined as active involvement in patients’ daily activities, medical follow-ups, treatment management, and addressing various needs. Participants were required to be family members, rather than professional caregivers, literate, and willing to voluntarily provide informed consent. Exclusion criteria included individuals with psychotic disorders, active mood episodes, intellectual disabilities, dementia, or other organic mental disorders, as well as any self-reported psychiatric disorder currently under treatment. This was determined via self-report, where participants were asked if they were currently receiving professional treatment for any diagnosed psychiatric condition. Individuals who were unwilling to consent or otherwise failed to meet the inclusion criteria were also excluded.

### 2.2. Data Collection

Data were collected from a total of 819 caregivers providing care to patients in palliative settings. This included 412 caregivers for cancer patients recruited from multiple hospitals across Turkey, and 407 caregivers for patients with chronic diseases, all undergoing clinical follow-up at the designated home health service clinic in Istanbul Tuzla State Hospital. The data collection spanned from October 2023 to September 2024, ensuring a broad and diverse sample of caregivers. Caregivers were recruited during routine visits for medical prescriptions or when applying to obtain a medical board report for disabled social rights. In an effort to circumvent bias, surveys were administered and completed under the supervision of volunteer personnel specifically trained for the study procedure, who were unaware of the patient’s diagnosis.

As patients and their caregivers applied to the clinic during the specified period, their data were sequentially recorded and evaluated according to the inclusion and exclusion criteria. Throughout this data collection process, interactions with the caregivers of the patients were conducted within the framework of privacy principles and ethical norms, involving informative and consent-obtaining procedures.

In the cohort of caregivers for cancer patients, the following exclusion criteria were applied, with corresponding numbers excluded: age over 65 years (*n* = 9), caregiving duration of less than six months following a new cancer diagnosis (*n* = 14), not undergoing active chemotherapy or not in a palliative care stage (*n* = 6), coexisting psychiatric disorders under treatment (*n* = 31), lack of consent to participate (*n* = 11), and inability to complete the assessment tools (*n* = 17). For caregivers of chronic disease patients, exclusions included: age over 65 years (*n* = 17), caregiving duration of less than six months (*n* = 10), coexisting psychiatric disorders under treatment (*n* = 46), lack of consent (*n* = 8), and inability to complete the assessment tools (*n* = 12).

A total of 500 participants per group were initially recruited, as determined by power analysis, to achieve sufficient statistical power. Recruitment concluded once this target was reached. Following data collection, retrospective power analyses were performed to validate the robustness and reliability of our findings concerning caregiver burden and psychosocial impact. After applying the predefined exclusion criteria, the final sample sizes of 407 and 412 participants per group were retained, ensuring statistical power exceeded 80% with a 5% margin of error. This post hoc validation confirmed that our study was adequately powered to detect significant effects and enhanced the reliability and validity of our results. These findings provide a robust foundation for understanding the multifaceted experiences and challenges encountered by caregivers in our selected cohorts.

The tools deployed for this data collection phase comprised a sociodemographic data form, the Zarit Care Burden Scale (ZCBS), Depression–Anxiety–Stress Scale (DASS-21), and the Relationship Scale Questionnaire (RSQ).

Participants were approached face-to-face during their visit. Participants were meticulously briefed on the study’s objectives, the procedural roadmap, alongside the potential benefits and risks before initiating the data collection, succeeded by acquiring their written informed consent. Each participant took approximately 30–45 min to complete the entire set of questionnaires.

Zarit Care Burden Scale (ZCBS): The ZCBS is a 22-item instrument designed to assess the burden experienced by informal caregivers. Each item is rated on a 5-point Likert scale ranging from 0 (never) to 4 (almost always), with higher total scores indicating greater caregiver burden. The Turkish adaptation of the ZCBS has shown good psychometric properties, with a reported Cronbach’s alpha of 0.83 [26]. In the current study, the scale demonstrated excellent internal consistency, with a Cronbach’s α = 0.913.

Depression–Anxiety–Stress Scale (DASS-21): The DASS-21 is a 21-item self-report instrument that evaluates depression, anxiety, and stress over the past week. Items are scored on a 4-point Likert scale reflecting the severity of symptoms. The Turkish version of the DASS-21 has demonstrated strong psychometric performance, with reported Cronbach’s α values of 0.87 (depression), 0.85 (anxiety), and 0.81 (stress) in clinical samples [27]. In the present study, internal consistency coefficients were α = 0.85 for depression, α = 0.81 for anxiety, and α = 0.94 for stress.

Relationship Scale Questionnaire (RSQ): The RSQ was developed by Griffin and Bartholomew [28] and adapted into Turkish by Sumer and Gungor [29]. It is a 30-item self-report instrument designed to assess four adult attachment styles: secure, fearful, preoccupied, and dismissing. Participants rate each item on a 7-point Likert scale, indicating how well each statement describes their typical relational patterns. The scale’s test–retest reliability coefficients range from 0.54 to 0.78, and its parallel form validity, assessed against the Relationship Questionnaire, yielded correlations between 0.49 and 0.61. In the original Turkish validation, Cronbach’s α values were reported as 0.81 (secure), 0.84 (fearful), 0.78 (preoccupied), and 0.79 (dismissing) [30]. In the current study, we treated the four attachment dimensions separately and found reliability coefficients of α = 0.81 (secure), α = 0.73 (fearful), α = 0.86 (preoccupied), and α = 0.76 (dismissing).

### 2.3. Ethical Considerations

The Ethics Committee of Istanbul Bilgi University (protocol code 2022-40162-156 and date of approval 7 December 2022) approved this study. It is performed in accordance with the Declaration of Helsinki. All patients provided written informed consent.

### 2.4. Statistical Analysis

Data analysis was conducted using SPSS 22 software (IBM Corp., Armonk, NY, USA). Distribution normality was assessed through skewness and kurtosis values, supplemented by the Kolmogorov–Smirnov test. To examine differences in scale scores across cancer patient subgroups categorized by cancer type, we utilized the Kruskal–Wallis test. Pearson’s correlation tests were applied based on data distribution to explore interrelationships between variables. Univariate and multivariate linear regression analyses were employed to identify independent determinants of caregiver burden among caregivers of palliative care cancer patients and those assisting chronic disease patients. To ensure the robustness of our regression models and mitigate potential multicollinearity, correlation coefficients between independent variables were examined, with no values exceeding the threshold of 0.80. Furthermore, Variance Inflation Factors (VIF) were assessed, remaining consistently below the accepted cutoffs of 3 or 10. Specifically, VIF values ranged from 1.032 to 2.080 for cancer caregivers and from 1.028 to 2.416 for chronic disease caregivers, confirming the absence of significant multicollinearity and substantiating the reliability of our models. To investigate potential moderation effects, we utilized the PROCESS macro for SPSS, available at [https://processmacro.org/index.html (accessed on 15 October 2024)] [31]. Moderation models were employed to explore the intricate relationships among depression, stress, attachment styles, and caregiver burden, as measured by the Zarit Caregiver Burden Scale. These analyses examined how the relationship between independent variables (depression or stress) and the dependent variable (caregiver burden) could be influenced by a moderator (attachment styles), potentially altering the strength or direction of the primary relationships. For cancer caregivers, Hayes Model 2 was applied to assess how specific attachment styles—secure and preoccupied—moderated the relationship between depression and caregiver burden. We hypothesized that these attachment styles could either attenuate or amplify the impact of depressive symptoms on caregiver burden. For chronic disease caregivers, Hayes Model 1 was utilized to investigate the moderating effects of attachment styles on the relationships between depression and caregiver burden, as well as stress and caregiver burden. This approach allowed us to determine whether attachment styles intensified or mitigated the impact of these psychological factors on caregiver burden. Initially, Hayes Model 1 was applied across all combinations of depression, anxiety, stress, and caregiver burden to assess moderation effects of all attachment styles. Significant relationships—such as the depression-caregiver burden relationship among cancer caregivers moderated by secure and preoccupied attachment—were subsequently analyzed using Hayes Model 2 for more detailed examination. To evaluate these relationships, we employed the bootstrap method with 5000 resamples, a non-parametric resampling strategy that does not require the assumption of normality. Bias-corrected 95% confidence intervals (CI) were used to interpret the moderation effects. Statistical significance was set at *p* < 0.05. The analysis was performed using a complete-case (or listwise deletion) approach. To maintain data integrity, only participants who fully completed all questionnaires were included in the final dataset. As detailed in the Data Collection section, this resulted in the exclusion of 29 participants (17 cancer caregivers, 12 chronic disease caregivers) who were unable to complete the assessments. The study was prepared according to the STROBE-Statement guideline recommended for cross-sectional studies, from Appendix A.

## 3. Results

The study encompassed 819 caregivers split evenly between those caring for individuals with cancer and those caring for individuals with chronic diseases. In the group of caregivers for cancer patients, the majority were providing care to individuals with breast cancer (33%), followed by lung (25.7%), colorectal (15.1%), pancreatic (11.2%), and gastric cancers (8.7%), with a minority attending to individuals with other types of cancers (6.3%). In our caregiver cohort for chronic patients, a substantial portion was dedicated to individuals with Alzheimer’s (46.0%), followed by those aiding stroke or cerebrovascular disease patients (37.3%); a smaller fraction of caregivers tended to individuals with intellectual disability (11.3%) and others (those physically disabled due to an accident, ALS and Parkinson’s) (9.3%). Their demographic details, including gender, education level, relationship to the patient, and for the cancer caregiver group, the type of cancer cared for, are illustrated below (Table 1).

Table 2 provides a comparative overview of the scores obtained from various scales—including the Zarit Caregiver Burden Score, the Relationship Scale Questionnaire (RSQ), and the Depression Anxiety Stress Scale-21 (DASS-21)—between two groups: caregivers tending to cancer patients and caregivers assisting individuals with chronic diseases. Caregivers of cancer patients reported significantly higher Zarit Caregiver Burden Scores, with a small-to-medium effect size (d = 0.35, 95% CI [0.21, 0.49]). They also scored higher on RSQ-Preoccupied Attachment (d = 0.55, 95% CI [0.41, 0.69]; medium effect), DASS-21 Anxiety (d = 0.47, 95% CI [0.33, 0.61]; small-to-medium effect), and DASS-21 Stress (d = 0.29, 95% CI [0.15, 0.43]; small effect). In contrast, chronic disease caregivers had significantly higher DASS-21 Depression scores, with a large effect size (d = –0.74, 95% CI [–0.88, –0.60]). No significant group differences were found for RSQ-Dismissing Attachment (d = 0.04, 95% CI [–0.10, 0.18]) or RSQ-Fearful Attachment (d = –0.10, 95% CI [–0.24, 0.04]). A small difference in RSQ-Secure Attachment was observed, favoring chronic disease caregivers (d = 0.18, 95% CI [0.04, 0.32]).

In the study cohort, the Zarit burden scores differed significantly across the various caregiver groups (*p* < 0.001). The group caregiving for pancreas cancer patients recorded the highest mean score at 33.02 ± 5.03 (*n* = 46), followed by those caring for gastric cancer patients with a mean score of 32.11 ± 4.48 (*n* = 38). Caregivers for colorectal cancer patients had a mean score of 30.11 ± 4.87 (*n* = 62), slightly above those caring for other cancer types, who reported a mean score of 29.96 ± 4.37 (*n* = 26). The groups caring for individuals with breast and lung cancers reported the lowest mean scores of 29.49 ± 4.23 (*n* = 136) and 29.52 ± 4.27 (*n* = 106), respectively. Conversely, in the context of chronic patient caregivers, the analysis did not indicate any significant disparity in the burden scores across different caregiver groups.

The secure attachment scores among caregivers tending to cancer patients were documented to vary based on the type of cancer diagnosed in the individuals they were caring for. The mean scores and their respective standard deviations for each group were as follows: breast cancer caregivers had a score of 4.31 ± 1.04 (*n* = 138), lung cancer caregivers recorded a score of 4.23 ± 0.83 (*n* = 106), and those caring for colorectal cancer patients had a score of 4.33 ± 0.84 (*n* = 62). Caregivers for gastric cancer patients reported a score of 4.12 ± 0.76 (*n* = 36), while those attending to individuals with pancreatic cancer had the lowest score of 3.93 ± 1.04 (*n* = 46). The group labeled ‘others’ exhibited a score of 4.32 ± 0.82 (*n* = 26). The overall mean score for the secure attachment across all groups was 4.23 ± 0.93 (*p* = 0.222).

Regarding the DASS-21 scales, cancer caregivers reported statistically significant lower scores for depression but higher scores for anxiety and stress.

Table 3 illustrates the relationships between different psychological measures, including various attachment styles and DASS-21 scores, with the Zarit Caregiver Burden Score in cancer caregivers, as demonstrated through Pearson correlation coefficients. The Zarit Caregiver Burden Score had a weak positive correlation with DASS-21 Depression (r = 0.338, *p* < 0.01) and very weak positive correlations with anxiety (r = 0.194, *p* < 0.01) and stress (r = 0.162, *p* < 0.01). A moderate negative correlation was observed between the Zarit score and secure attachment (r = −0.452, *p* < 0.01). Preoccupied attachment showed a weak positive correlation with burden (r = 0.276, *p* < 0.01), while fearful attachment had a very weak positive correlation (r = 0.125, *p* < 0.05).

Table 4 presents the correlation coefficients depicting the relationships between the Zarit Caregiver Burden Score and several psychological variables, including different attachment styles and DASS-21 subscales, for caregivers of individuals with chronic diseases. The Zarit Caregiver Burden Score demonstrated very weak positive correlations with DASS-21 Depression (r = 0.198, *p* < 0.01), Anxiety (r = 0.153, *p* < 0.01), and Stress (r = 0.118, *p* < 0.05). Secure attachment had a very weak negative correlation with caregiver burden (r = −0.149, *p* < 0.01). Both preoccupied (r = 0.169, *p* < 0.01) and fearful attachment styles (r = 0.137, *p* < 0.01) also showed very weak positive correlations with the Zarit Caregiver Burden Score.

Table 5 and Table 6 delineate the results of the univariate and multivariate regression analyses, which elucidate the relationships between the Zarit Caregiver Burden Score and various factors. These factors include psychological metrics such as secure, dismissing, preoccupied, and fearful attachments, along with other demographic and caregiving-specific variables in cancer caregivers and caregivers of individuals with chronic diseases.

The regression model, which included age, gender, relationship to the patient, caregiving duration, type of malignancy, DASS-21 subscores for depression, anxiety, and stress, and attachment types—secure, preoccupied, and fearful—as independent variables, explained 43.1% of the variance in the Zarit Caregiver Burden Score (*F(11,400)* = 29.312, *p* < 0.001). All variables, except for age and fearful attachment, significantly predicted Zarit Caregiver Burden Scale scores independently. Secure attachment demonstrated a substantial negative correlation with the Zarit score, indicating a reduction in caregiver burden with increased secure attachment. DASS-21 scores for depression, anxiety, and stress were positively associated with the Zarit score in both univariate and multivariate analyses, reflecting a rise in caregiver burden with higher levels of these factors. Additionally, preoccupied attachment showed a significant positive association with the Zarit score in both analyses, suggesting increased caregiver burden with greater preoccupied attachment tendencies. Notably, while age and fearful attachment were significant in the univariate analysis, they did not retain significance in the multivariate model (Table 5).

To assess the robustness of the model, we examined multicollinearity and serial correlation. Variance Inflation Factors (VIFs) for all predictors were below 2.1, indicating no multicollinearity concerns. The highest VIF observed was for relationship to patient (VIF = 2.039), followed by duration of caregiving (VIF = 1.169) and type of cancer (common/non-common) (VIF = 1.104). The Durbin–Watson statistic was 2.082, suggesting that residuals were not serially correlated. Finally, analysis of influential observations showed that Cook’s Distance values were all below 0.171, and leverage and studentized residual diagnostics indicated no influential outliers, confirming the stability of the model.

The regression model, which included age, gender, DASS-21 subscores for depression, anxiety, and stress, and attachment types—secure, preoccupied, and fearful—explained 10.9% of the variance in the Zarit Caregiver Burden Score (*F(8, 398*) = 7.188, *p* < 0.001). Gender and depression scores showed the strongest positive association with the Zarit score, indicating a marked increase in caregiver burden with higher levels of depression and being male. Preoccupied attachment and stress scores were also positively associated with the Zarit score in the univariate analysis, suggesting increased caregiver burden with greater preoccupied attachment tendencies and higher stress levels; however, these associations did not remain significant in the multivariate analysis. Age was negatively associated with Zarit burden scores in the univariate analysis, but this association was not significant in the multivariate model. Secure attachment showed a negative association with the Zarit score in the univariate analysis, suggesting that higher secure attachment might reduce caregiver burden, though this relationship also did not retain significance in the multivariate analysis (Table 6).

To assess the integrity of the model, collinearity statistics were examined. All predictors had VIF values below 2.5, indicating no multicollinearity concerns. The highest VIF was observed for secure attachment (VIF = 2.416), followed by fearful attachment (VIF = 1.701) and preoccupied attachment (VIF = 1.672). Additionally, the Durbin–Watson statistic was 1.825, suggesting no serial correlation in residuals. Influential case diagnostics, including Cook’s Distance (max = 0.048) and studentized deleted residuals (±2.58), showed no concerning outliers or undue influence on model estimates.

Investigations into the roles of various attachment styles—secure, fearful, preoccupied, and avoidant—were conducted for both caregiver groups, cancer caregivers and chronic disease caregivers, through several moderation analyses. Bridging these explorative methodologies with our targeted analysis, the following significant moderations were uncovered:

In our analysis deploying Hayes process analysis model 2, we examined the moderator role of secure and preoccupied attachment in the relationship between depression (as the independent variable) and caregiver burden among cancer caregivers measured through the Zarit scale (as the dependent variable). Upon assessing individual predictors, we observed a statistically significant main effect of depression on caregiver burden (B = 1.303, SE = 0.141, t = 9.227, *p* < 0.001), indicating that higher levels of depression are associated with increased caregiver burden. Secure attachment showed a significant negative association with caregiver burden (B = −2.032, SE = 0.193, t = −10.537, *p* < 0.001), suggesting that higher levels of secure attachment are linked to a reduction in caregiver burden.

The interaction between depression and secure attachment was significant (B = −0.611, SE = 0.127, t = −4.804, *p* < 0.001), indicating that the association between depression and caregiver burden is moderated by secure attachment. Specifically, as secure attachment increases, the impact of depression on caregiver burden appears to decrease, suggesting a buffering effect of secure attachment on the relationship between depression and caregiver burden.

Preoccupied attachment also demonstrated a significant positive main effect on caregiver burden (B = 0.800, SE = 0.161, t = 4.978, *p* < 0.001), indicating that higher levels of preoccupied attachment are associated with greater caregiver burden. Additionally, the interaction term between depression and preoccupied attachment (Int_2) was significant (B = 0.292, SE = 0.116, t = 2.508, *p* = 0.013), suggesting that the relationship between depression and caregiver burden is also moderated by preoccupied attachment. In this case, higher levels of preoccupied attachment amplify the effect of depression on caregiver burden. Figure 1 illustrates the conditional effects of secure and preoccupied attachment on the depression-caregiver burden relationship among cancer caregivers at different attachment levels (mean-SD, mean, mean + SD). The red line represents low secure attachment, the blue line represents mean secure attachment, and the green line represents high secure attachment.

We examined the moderating role of preoccupied attachment on the relationship between depression and caregiver burden among chronic disease caregivers. Depression was positively associated with caregiver burden (β = 0.773, SE = 0.226, t = 3.424, *p* = 0.001), suggesting that higher levels of depression correlate with increased caregiver burden. Preoccupied attachment was also positively associated with caregiver burden (β = 0.699, SE = 0.315, t = 2.223, *p* = 0.027), indicating that caregivers with higher levels of preoccupied attachment tend to experience greater caregiver burden.

Central to our analysis was the interaction between depression and preoccupied attachment, which was statistically significant (β = 0.505, SE = 0.169, t = 2.986, *p* = 0.003), suggesting a moderating role of preoccupied attachment in the depression–caregiver burden relationship. The findings reveal that as levels of preoccupied attachment increase, depression has a progressively greater impact on caregiver burden. This relationship is statistically significant at mean and high levels of preoccupied attachment, but not at low levels. Figure 2 illustrates the conditional effects of preoccupied attachment on the depression–caregiver burden relationship among chronic disease caregivers at different attachment levels (mean, mean + SD). The black line represents high preoccupied attachment, while the gray line represents mean preoccupied attachment.

In our examination of the moderating impact of preoccupied attachment on the relationship between stress and caregiver burden among chronic disease caregivers, we found that stress was positively associated with caregiver burden (β = 0.219, SE = 0.105, t = 2.082, *p* = 0.038), indicating that higher levels of stress correlate with increased caregiver burden. Preoccupied attachment was also positively associated with caregiver burden (β = 0.881, SE = 0.316, t = 2.789, *p* = 0.006), suggesting that caregivers with higher levels of preoccupied attachment tend to experience greater burden.

Notably, the interaction between stress and preoccupied attachment was significant (β = 0.199, SE = 0.089, t = 2.237, *p* = 0.026), indicating a meaningful moderating effect. The findings reveal that as preoccupied attachment increases, it amplifies the adverse impact of stress on caregiver burden. This moderating effect is statistically significant at mean and high levels of preoccupied attachment, but not at low levels. Figure 3 illustrates the conditional effects of preoccupied attachment on the stress–caregiver burden relationship among chronic disease caregivers at different attachment levels (mean, mean + SD). The black line represents high preoccupied attachment, while the gray line represents mean preoccupied attachment.

## 4. Discussion

In our study with a cohort of 819 caregivers, we identified significant demographic and psychological differences between caregivers of cancer patients and those caring for patients with chronic diseases. The DASS-21 scales indicated that, while cancer caregivers exhibited lower depression scores, they reported higher levels of anxiety and stress compared to chronic disease caregivers. Furthermore, correlation analyses using the Zarit Caregiver Burden Score revealed that caregiver burden was positively associated with DASS-21 depression, anxiety, and stress scores in both groups, while secure attachment was negatively correlated with caregiver burden. The regression analysis explained 43.1% of the variance in caregiver burden among cancer caregivers and 10.9% among chronic disease caregivers. Significant predictors for cancer caregivers included male gender, being the patient’s child, caregiving for more than one year, secure and preoccupied attachments, as well as DASS-21 depression, anxiety, and stress. For chronic disease caregivers, significant predictors included depression, anxiety, fearful attachment, and male gender. Secure and preoccupied attachment were found to significantly moderate the relationship between depression and caregiver burden among cancer caregivers. Specifically, higher levels of depression were associated with increased caregiver burden, but secure attachment appeared to mitigate this effect, while preoccupied attachment amplified it. Our findings further demonstrated that preoccupied attachment significantly moderates both the depression–caregiver burden and stress–caregiver burden relationships among chronic disease caregivers, intensifying the impact of depression and stress on caregiver burden as levels of preoccupied attachment increase. These results highlight the complex psychological landscape of caregiving and the critical need for tailored caregiving strategies that address the unique challenges faced by each group. It is essential to emphasize the substantial emotional and psychological burdens borne by caregivers, with distinct challenges depending on whether they are supporting individuals with cancer or chronic diseases.

The data presented elucidates a complex emotional landscape that these caregivers navigate daily. From the beginning, cancer caregivers grapple with the acute crises synonymous with cancer diagnoses, often accompanying patients through traumatic junctures of pain, suffering, and an overwhelming sense of impending loss [32,33,34,35]. The palpable sense of urgency and despair in their caregiving journey is reflected in the higher Zarit Caregiver Burden Score as compared to those caring for individuals with chronic diseases. Comparatively, ZBI scores spotlight the wide-ranging hardships encountered by caregivers in disparate medical contexts. For instance, Alzheimer’s caregivers in China have reported a mean ZBI score of 12.2 ± 13.2, revealing not only significant caregiver burden but also concurrent psychological strains, such as depression and anxiety [36]. Conversely, Hooley et al. uncovered a ZBI score of 16.0 ± 14.4 among caregivers of congestive heart failure patients, indicating a complex interplay between caregiver depressive symptoms and the severity of the patient’s condition [37]. Moreover, dealing with Parkinson-related dementia appears notably taxing, with caregivers experiencing a substantially high average ZBI score of 35.51, as documented by Vatter et al. in 2018 [38]. These diverse caregiving contexts underscore the variable yet universally impactful nature of caregiver burden, as reflected through ZBI scores, irrespective of the specific medical condition at hand. Drawing parallels to our findings, the study conducted by Burton et al. delineated the burden and well-being among a diverse cohort of caregivers, nursing individuals grappling with cancer, congestive heart failure, and chronic obstructive pulmonary disease. Interestingly, they reported no differences in the levels of depressive symptoms, anxiety, and spiritual well-being across different caregiver groups, steering us towards a contemplation of the unifying threads in the caregiver narrative, regardless of the specific ailment at hand [39]. The higher scores in the preoccupied attachment dimensions may indicate a heightened sense of obsession and an ever-present fear of loss, portraying a caregiving experience rife with emotional turbulence and distressing unpredictability. The DASS-21 scores accentuate the substantial stress levels these caregivers encounter, potentially driven by a ceaseless quest to find solutions in palliative care, mitigating the unbearable, and often ambiguous, pathway that cancer prognoses often delineate.

In contrast, caregivers attending to chronic disease patients often find themselves immersed in a protracted caregiving trajectory, a journey that not only extends over a substantial period but is punctuated with sustained apprehensions tied to the gradual deterioration of the patient’s health status. According to Lambert, Levesque, and Girgis, caregiving, especially in the context of cancer patients, plays a pivotal role in enhancing the patients’ well-being while alleviating the pressure on the healthcare infrastructure. However, this comes at the cost of caregivers potentially experiencing heightened burden and diminished quality of life, translating to a profound psychological impact over time [40]. Reflecting on our findings, it appears that the caregivers in this group exhibited elevated depression scores on the DASS-21. This could possibly hint at an accrual of psychological distress over time, a cumulative burden spawned from the continual witness to the slow but steady decline in the patient’s health, fostering a backdrop of anticipatory grief and perennial worry for the loved ones under their care. Interestingly, our study noted diminished scores in the preoccupied attachment style, a potential indication of caregivers adapting to a pace of care that, while inexorable, intersperses moments of stability and predictability within a framework of continual concern. To delve deeper, future explorations could aim to unravel the nuanced dynamics at play, offering a more rounded perspective on the oscillating phases of stability and anxiety that characterize the chronic disease caregiving landscape, thus illuminating pathways for more tailored and effective support mechanisms.

A more granular examination of the correlation strengths provides a deeper insight into the distinct nature of caregiver burden in our two cohorts. For cancer caregivers, the analysis revealed that secure attachment had a moderate negative relationship with burden (r = −0.452), while depression had a weak positive relationship (r = 0.338). Notably, the direct correlations of anxiety, stress, and other attachment styles were very weak. This suggests that for caregivers facing the acute crisis of palliative cancer care, their core relational security (attachment style) and mood (depression) are more substantively linked to their experience of burden than the more situational feelings of anxiety or stress. The moderate strength of secure attachment’s effect is particularly noteworthy, highlighting its role as a robust protective factor in this high-stakes environment.

In stark contrast, a critical finding for chronic disease caregivers was that all significant correlations between the measured psychological variables and caregiver burden were very weak. This pattern strongly suggests that while factors like depression and attachment styles are statistically significant [41], they may not be the primary drivers of burden in long-term caregiving situations. This aligns with our regression model for this group, which explained only a small fraction of the variance (10.9%). The weakness of these associations implies that the burden for chronic disease caregivers is likely dominated by other unmeasured variables. These could include the chronicity of care, the immense physical toll over years, long-term financial strain, and social support erosion—factors that accumulate over time and may overshadow the influence of the specific psychological traits measured here. This finding underscores that a different theoretical model may be needed to fully capture the experience of burden in chronic care.

The preoccupied attachment style emerged as a prominent factor, registering a positive correlation with the Zarit score. This correlation possibly mirrors a caregiving milieu where caregivers are persistently apprehensive, veering towards over-involvement to an extent that manifests as substantial burden. This synergy with the traits of a preoccupied attachment style is illustrated vividly in existing literature, painting a picture of caregivers who are excessively involved, characterized by a proclivity towards worry and a tendency to engage in compulsive caregiving. This scenario crafts a canvas of mutual exhaustion, delineating a space where both the caregiver and the patient navigate a complex labyrinth of heightened emotions and perpetual concerns. Reinforcing this, Nicholls et al. noted a significant association between more insecure attachment styles, including the preoccupied attachment style, and a myriad of challenges such as escalated caregiving stress, depressive symptoms, less autonomous motivations for caregiving, and substantial difficulties navigating the caregiving journey. This intersection of heightened involvement and increasing stressors underscores a critical avenue for further research, hinting at the necessity to foster secure attachment dynamics to potentially ameliorate the psychological distress intertwined with caregiving roles, thereby nurturing a healthier caregiver-patient dynamic [42]. Moreover, the caregivers with preoccupied attachment style may find themselves relentlessly pursuing the optimal solution for the patient, endeavoring to provide the finest care possible. This unyielding commitment not only intensifies the emotional and psychological toll on them but also solidifies the evident manifestation of the preoccupied attachment style in this context. The accompanying burden is reflected markedly in the Zarit score, a testament to the complex dynamics of the caregiver-patient relationship in chronic disease settings. Furthermore, this lens clarifies the role of preoccupied attachment. While its direct correlation with burden was weak or very weak, our moderation analysis showed it significantly amplified the effects of depression and stress. This implies that preoccupied attachment may function less as a direct cause of burden and more as a vulnerability factor, a psychological sensitizer that makes caregivers more susceptible to the negative effects of other stressors, rather than being a major source of burden in itself.

Table 5 and Table 6 unfurl a complex interplay of psychological and demographic factors in the caregiving setting, significantly influenced by variables such as attachment styles, stress levels, anxiety and depression. A key insight from the regression analysis in Table 5 underscores the pivotal, yet multifaceted, role of secure attachment in mitigating caregiver burden, correlating with reduced Zarit scores and acting as a protective buffer against them [42]. Not only does secure attachment foster resilience and enhance stress management, thereby serving as a potential therapeutic avenue in mitigating caregiver distress, but it also aids caregivers in realistically foreseeing and preparing for potential loss, especially pertinent for caregivers tending to individuals with severe health conditions like cancer.

In a clinical context, caregivers without secure attachments often delve into unmet expectations, attempting to fulfill a lifetime of dreams in a constrained timeline and inadvertently intensifying their caregiving burden. The lack of secure attachment, therefore, can initiate an exhaustive spiral, emphasizing its importance not only as a support during caregiving but also as a facilitator of understanding and acceptance in navigating through impending loss [43,44]. Furthermore, the DASS-21 stress metrics illuminate how escalated stress levels can markedly amplify caregiver burden, accentuating the crucial role of targeted stress management interventions as proactive measures to mitigate the intensifying burden inherent in the caregiving role. While age emerged as a significant factor in the univariate analysis, its predictive strength diminished in the multivariate landscape, indicating a potential, more complex interaction with other demographic and psychological factors that merits further investigation.

Caregiving for prevalent cancer types, notably breast and lung cancers, was associated with a protective effect in terms of burden in the univariate analysis [45,46,47]. This could be linked to numerous factors, such as well-established care pathways, which might provide structured guidance, and the existence of substantial support communities offering a robust network of peers and resources. Moreover, the ample availability of information on these common cancers might equip caregivers with knowledge that mitigates the sense of uncertainty, while the familiarity with these cancers could potentially lead to perceptions of predictability and controllability, thereby attenuating caregiver burden.

Turning to Table 6, it is evident that higher levels of depression are associated with increased caregiver burden. This aligns with a substantial body of literature emphasizing the cycle of caregiving and depression, underscoring the urgent need for integrated mental health support for caregivers [48,49,50]. The finding regarding fearful attachment further emphasizes its strong positive association with the Zarit score, suggesting a potentially exhausting caregiving experience marked by heightened anxiety and apprehension. This aligns well with the complexities outlined in fearful attachment theory, highlighting the need for intervention plans that consider diverse attachment styles with a nuanced approach. In the univariate analysis, secure attachment showed a negative correlation with caregiver burden; however, this protective effect did not persist in the multivariate analysis, inviting deeper exploration into the interplay of variables in a multivariate context. This shift around secure attachment raises important questions about its role and presents a promising area for future research.

The observed potential higher risk of caregiver burden among males may be rooted in sociocultural norms and psychological frameworks, highlighting traditional gender roles and the uncommon expectation for males to take on caregiving roles compared to females. This discrepancy suggests that males might employ different strategies to handle caregiver stress, possibly affecting their experience of caregiver burden distinctly. These findings signal a necessity for detailed research into gender disparities in caregiving, aiming to unravel gender-specific experiences and thereby fostering more tailored support systems.

Exploring the complexities of caregiver-patient interactions, especially in the landscapes of cancer and chronic disease, our research harmoniously aligns with and subtly expands the existing literature, solidifying the concept that attachment styles are instrumental in shaping caregiver-patient interactions and caregivers’ psychosocial experiences. Nowadays, research increasingly seeks to identify moderators and mediators of psychological factors influencing caregiver burden, as seen in Cheng’ study, which highlights self-esteem and psychological distress as mediators between caregiver burden and quality of life among caregivers of individuals with severe mental illness [51]. Our investigation unveiled distinct roles of various attachment styles in modulating the relationships between psychological stresses and caregiver burden. Remarkably, a heightened sense of secure attachment emerged as a mitigating force, dampening the repercussions of depression on caregiver burden, and acting as a potential cushion that absorbs the strains of depression, thereby forming a protective barrier against escalating caregiver burden.

Parallelly, our findings resonate with those of Nissen and Vachon, illuminating a stark contrast between secure and insecure attachments in managing emotional strain and caregiving efficacy, respectively. Notably, avoidant attachment style did not significantly dictate caregiver burden, hinting at a possible protective role through emotional distancing, as suggested in the previous literature [52]. This insinuates that even avoidant strategies might offer a form of coping, pointing to a rich and complex landscape of caregiving dynamics that merits further exploration. Moreover, our observations of the non-significant influence of avoidant attachment on caregiver burden hint at coping mechanisms through emotional distancing, an aspect somewhat corroborated by Nicholls et al. but needs deeper exploration in future studies [42]. On a separate note, while our exploration into gender dynamics echoes themes from Tsilika et al., it underscores that existing literature may not delve as deeply into comprehending how societal expectations shape caregiving experiences across different attachment styles [25].

Integrating these insights, it becomes pivotal to recognize the crucial role of secure attachment—not only in enhancing caregiver experiences, as echoed by findings from Karveli et al., but also in shaping comprehensive caregiver support mechanisms [53]. Additionally, the intensifying effect of preoccupied attachment is noteworthy: among cancer caregivers, it amplifies the impact of depression on caregiver burden, while among chronic disease caregivers, it heightens both the stress-caregiver burden and depression-caregiver burden relationships. Consequently, our future trajectory necessitates a more nuanced, integrated exploration into caregiving, one that intricately weaves together attachment styles with the emotional and physical dimensions of caregiving. This approach aims to foster a caregiving model that is both emotionally supportive and practically effective amidst the multifaceted challenges of chronic and cancer care.

### 4.1. Limitations

While this study provides valuable insights into caregiving dynamics, several limitations should be acknowledged. Conducted in Turkey, a country with strong familial ties and a cultural emphasis on family caregiving, our findings are deeply rooted in this socio-cultural context. This specificity may limit the global applicability of our results, as socio-cultural norms in Turkey might influence attachment styles and caregiving experiences in ways that differ from other cultural settings. Future studies encompassing diverse cultural contexts are needed to establish the universality of these findings. Although the study sample size was adequate, larger and multinational cohorts are required to enhance the generalizability and granularity of the observed trends. A broader participant pool may uncover subtler dynamics and associations that were not evident in our sample. Moreover, the cross-sectional design, while offering a snapshot of caregiving experiences, restricts the ability to establish causality between attachment styles, psychological stress, and caregiver burden. This limitation confines our understanding to a single time point, potentially overlooking the evolving nature of caregiving stressors and experiences over time. Longitudinal studies are warranted to explore these dynamics more comprehensively. Focusing specifically on caregivers of palliative cancer patients and chronic disease patients receiving home health services provided specialized insights but limited the generalizability of our findings to other caregiver populations. These results should be interpreted cautiously when applied to broader caregiving contexts or other patient groups, as different caregiving scenarios may yield distinct challenges and burdens. Furthermore, our restrictive inclusion criteria for the cancer cohort—specifically focusing on caregivers of patients with stage 4 solid tumors no longer receiving chemotherapy—limits the generalizability of our findings to caregivers at other stages of the cancer trajectory or during active treatment. Similarly, the exclusion of caregivers over 65 years of age may not capture the experiences of a significant and often vulnerable segment of the caregiving population. Finally, the chronic disease cohort was intentionally broad, encompassing conditions such as Alzheimer’s disease, stroke, and intellectual disabilities. While this provides a wide-ranging view, this heterogeneity is a limitation, as the unique psychological demands of each condition may be masked. This asymmetry between a narrowly defined, end-of-life cancer cohort and a highly heterogeneous chronic disease cohort may also impact the direct comparability of the two groups, as the observed differences in burden could be attributable to disease trajectory and acuity as much as the cancer versus chronic disease distinction itself. Future research should aim to disaggregate these groups to identify disease-specific caregiver dynamics. Our recruitment method, a form of convenience sampling that targeted caregivers already visiting medical clinics, introduces a potential selection bias. This approach inherently excludes caregivers who may be more isolated or do not regularly access hospital services, and their experiences may differ significantly from our study population. Additionally, while we specified a minimum caregiving duration of six months, the assessment was not standardized to a specific point in the patient’s illness trajectory. Given that caregiver burden and psychological distress can fluctuate greatly over time, this lack of specific timing is a limitation in interpreting the reported experiences. The inclusion criterion requiring literacy may have introduced a socio-economic selection bias, potentially excluding older or rural caregivers who represent a significant portion of the caregiving population in some contexts. This limits the generalizability of our findings to these specific demographics. Future research should consider employing interviewer-administered questionnaires to ensure the inclusion of non-literate participants. The reliance on self-reported data introduces the possibility of response and recall biases, potentially influencing the accuracy and objectivity of the findings. Participants’ subjectivities may subtly shade the results, affecting the reliability of the reported associations. Furthermore, while the use of trained volunteers aimed to standardize the data collection process, the procedural consistency among these volunteers was not formally assessed. Subtle variations in how the questionnaires were administered could have introduced unintended variability into the data, representing a minor methodological limitation. Future studies could strengthen internal validity by implementing periodic calibration meetings for data collectors. The omission of potentially influential confounding factors represents another limitation. Variables such as the caregiver’s own physical health, the availability of social support, their financial status, the number of hours spent caregiving daily, and their level of health literacy or any formal caregiving training they may have received could act as primary drivers of burden, potentially overriding or interacting with the effects of attachment style. For instance, a caregiver with a strong support network might experience low burden regardless of their attachment style, which is a dynamic our model could not capture.

Our study did not formally assess caregivers’ health literacy or the impact of any standardized training. While caregivers received basic medical information about the patient’s disease progression and management during clinical visits, this was not a structured intervention aimed at managing caregiver burden itself. Therefore, the effect of targeted caregiver education, alongside other factors like the caregiver’s own physical health, social support, and financial status, remain important unmeasured variables. Additionally, while we excluded caregivers with a self-reported psychiatric disorder under treatment, we did not screen for undiagnosed conditions. It is therefore likely that the psychological distress captured by the DASS-21 reflects a spectrum of symptoms among participants, some of whom may have had untreated mood or anxiety disorders. This exclusion criterion, while standardizing the sample, may have introduced a selection bias by excluding individuals actively seeking mental health support. At the same time, excluding caregivers with diagnosed psychiatric or cognitive disorders limits the ‘real-world’ generalizability of our findings, as a substantial portion of caregivers may experience such conditions Consequently, our results may not fully capture the complex interplay between pre-existing mental health challenges and the burdens of caregiving. Future research should seek to include these populations, perhaps using adapted assessment methods, to provide a more comprehensive understanding of caregiver burden. These limitations not only temper the applicability and interpretability of our findings but also illuminate pathways for future research, which should seek to expand, validate, and deepen the understanding established herein, thereby crafting a more global and comprehensive picture of caregiver experiences and burdens across varied contexts and demographics.

### 4.2. Clinical Implications

In managing the complexities of cancer care, caregivers stand as crucial pillars, shouldering the multifaceted demands of palliative care. Given the essential role of palliative care in enhancing the quality of life for patients with advanced cancer, understanding the dynamics of caregiver burden is paramount. Secure attachment styles have been identified as potential protective factors, shown to reduce caregiver burden. Thus, targeted interventions that foster secure attachment, particularly during the intensive palliative phases, may represent a promising avenue for improving caregiver well-being. Identifying caregivers with preoccupied attachment styles early on is also critical, as these styles are linked to heightened caregiver burden, particularly under stress and depressive symptoms. The strong positive correlation we identified between preoccupied attachment and caregiver burden extends the findings of previous studies and highlights a critical clinical challenge. This finding suggests that for caregivers with a preoccupied style, the act of caregiving may become a cycle of compulsive, overly involved behavior that, while well-intentioned, is psychologically detrimental. The clinical implication of this is significant: a substantial portion of caregivers may be at high risk for burnout not because of the patient’s needs alone, but because of an underlying relational pattern that drives them toward exhaustion. From a public health perspective, this has clear implications. Generic support systems that offer only practical aid, such as respite care or financial assistance, may be insufficient for this vulnerable group. Policy and healthcare systems should therefore consider a dual-support model. This would involve combining practical aid with easily accessible, targeted psychological programs—such as short-term counseling or guided support groups—designed to help these caregivers recognize these patterns, develop healthier boundaries, and improve self-efficacy. Such an approach could be a cost-effective strategy to prevent the long-term mental health crises associated with caregiver burnout. Tailored support for these caregivers could mitigate the strain reflected by elevated Zarit scores, which signify greater caregiver distress. Addressing the unique needs of caregivers based on gender can further personalize and enhance these interventions, ensuring more precise and effective support. Beyond the physical tasks of caregiving, the psychological dimensions, especially depression and stress, demand dedicated attention due to their clear association with caregiver burden. As noted by Otani et al., caregivers with insecure attachment styles, particularly those with heightened interpersonal sensitivity, are more prone to emotional exhaustion. This suggests that attachment-focused interventions may be crucial in helping caregivers navigate their emotional responses, thereby alleviating psychological distress and reducing overall Burden [54]. Embedding attachment-focused therapeutic interventions within palliative care settings may serve as a supportive measure against the detrimental effects of psychological stressors, offering a buffer for caregivers under strain. Future longitudinal and intervention studies are warranted to test these hypotheses and establish a causal relationship. The implementation of such support would likely involve a multi-disciplinary effort. Family doctors or Palliative care teams, including physicians and nurses, could be trained to screen for attachment-related distress during routine interactions. Following this initial screening, caregivers could then be referred to dedicated mental health professionals, such as clinical psychologists, social workers, or specialized counselors integrated within the care system, who are trained to deliver attachment-based interventions. These interventions could take various forms depending on the context and available resources, ranging from individual psychotherapy focused on exploring caregiver attachment patterns to therapist-led support groups that help normalize experiences and foster peer support, or structured psychoeducational programs designed to enhance secure coping strategies. The critical need for these tailored interventions is underscored by the profound impact of attachment style on a caregiver’s ability to manage emotional distress. Bianciardi et al. emphasized that insecure attachment styles can significantly impair emotional regulation in high-stress environments. This reinforces the need for structured emotional support programs tailored to caregivers who struggle with emotional regulation, ensuring they receive guidance in managing stress and anxiety effectively [55].

Additionally, Dell’Osso et al. highlighted that unresolved attachment insecurities contribute to heightened distress in the face of loss. Given that many cancer caregivers experience anticipatory grief, incorporating grief-focused interventions within caregiver support programs could help mitigate long-term emotional consequences, fostering resilience and adaptive coping strategies [56].

In conclusion, as we endeavor to bridge “cancer therapy” with the compassionate act of “caring,” an integrated approach is imperative. Such an approach must prioritize not only the physical well-being of patients but also the psychological resilience of caregivers, paving the way for a more holistic and empathetic palliative care paradigm.

## 5. Conclusions

To our knowledge, this is the first study in the literature to compare the cumulative influence of attachment styles, depression, stress, and anxiety on caregiver burden among cancer and chronic disease caregivers. Embarking on a comprehensive exploration into the profound interconnections between attachment styles and caregiver mental well-being, our study illuminates the substantial impact of these styles on the psychological trajectories of those aiding patients with cancer and chronic diseases. Navigating through unique relational dynamics, we have unveiled how distinct attachment styles not only shape the caregiving journey but also intricately mold the emotional and mental pathways of caregivers, adding an intricate layer to their multifaceted role. Our research extends beyond mere theoretical understanding, endeavoring to craft a nuanced framework that informs the development of practical, targeted caregiver support interventions. This framework, carved from our explorations, stands as a beacon to navigate caregiver support mechanisms toward recognizing and addressing the unique psychological and emotional challenges presented by different attachment styles. Our ultimate objective moves beyond understanding and into the realm of tangible support, aspiring to ensure caregivers are robustly understood, supported, and empowered throughout their complex caregiving journey, thereby enhancing resilience and ensuring a structured support network. Future endeavors must focus on translating these insights into actionable strategies, ensuring caregivers’ mental and emotional well-being is consistently championed and effectively supportd in both practice and policy.

## Figures and Tables

**Figure 1 healthcare-13-02612-f001:**
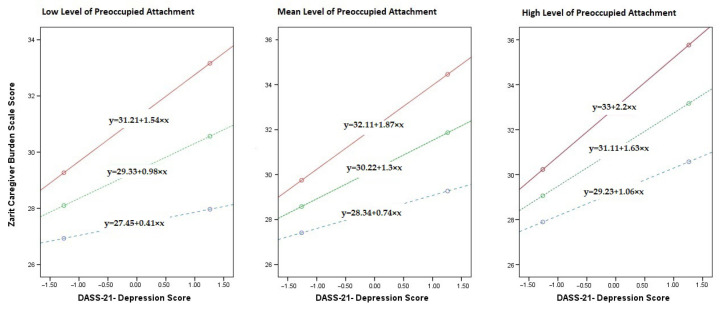
Secure and Preoccupied Attachment Effects on Depression–Caregiver Burden in Cancer Caregivers.

**Figure 2 healthcare-13-02612-f002:**
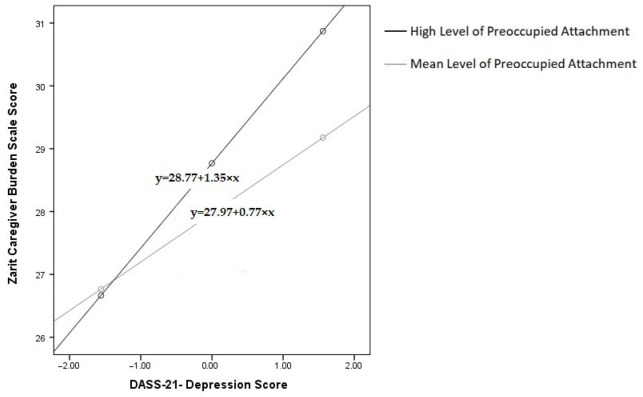
Preoccupied Attachment Effects on Depression–Caregiver Burden in Chronic Disease Caregivers.

**Figure 3 healthcare-13-02612-f003:**
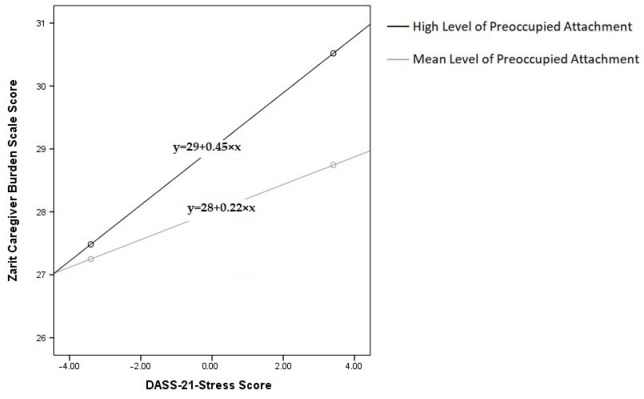
Preoccupied Attachment Effects on Stress–Caregiver Burden in Chronic Disease Caregivers.

**Table 1 healthcare-13-02612-t001:** Sociodemographic and Clinical Features.

		Cancer Caregivers (*n* = 412)	Chronic Disease Caregivers (*n* = 407)	*p* Value
	Age	44.85 ± 10.49	41.11 ± 8.91	**<0.001**
Gender	Female	245 (59.5)	264 (64.9)	0.111
	Male	167 (40.5)	143 (35.1)	
Marital status	Single/Divorced/Widow	99 (24.0)	78 (19.2)	0.091
	Married	313 (76.0)	329 (80.8)	
Education	Primary/ Secondary school	157 (38.1)	153 (37.6)	0.078
	High school	147 (35.7)	171 (42.0)	
	College/PhD	108 (26.2)	83 (20.4)	
Relationship	Children	124 (30.1)	159 (39.1)	**0.009**
	Spouse	145 (35.2)	108 (26.5)	
	Parent	55 (13.3)	43 (10.6)	
	Sibling	88 (21.4)	97 (23.8)	
Cancer type	Breast	136 (33.0)		
	Lung	106 (25.7)		
	Colorectal	62 (15.1)		
	Gastric	36 (8.7)		
	Pancreatic	46 (11.2)		
	Other *	26 (6.3)		

* Ovarian, testicular, bladder cancer; cholangiocarcinoma; Numbers indicate mean ± Standart Deviation or *n* (%).Values in bold represent statistically significant differences (*p* < 0.05).

**Table 2 healthcare-13-02612-t002:** Comparison of Scale Scores between Cancer Patients and Healthy Controls.

	Cancer Caregivers (*n* = 412)	Chronic Disease Caregivers (*n* = 407)	*p* Value
Zarit Caregiver Burden Sore	30.24 ± 4.60	28.12 ± 7.24	**<0.001**
RSQ-Secure Attachment	4.23 ± 0.93	4.06 ± 0.96	**0.007**
RSQ-Dismissing Attachment	2.83 ± 0.64	2.80 ± 0.88	0.582
RSQ-Preoccupied Attachment	3.91 ± 1.11	3.29 ± 1.14	**<0.001**
RSQ-Fearful Attachment	4.81 ± 1.22	4.94 ± 1.35	0.168
DASS-21-Depression	7.37 ± 1.26	8.42 ± 1.56	**<0.001**
DASS-21-Anxiety	6.23 ± 1.73	5.32 ± 2.10	**<0.001**
DASS-21-Stress	12.79 ± 3.45	11.79 ± 3.41	**<0.001**

Numbers indicate mean ± Standard Deviation; Values in bold represent statistically significant differences (*p* < 0.05); DASS-21, Depression Anxiety Stress Scale-21; RSQ, Relationship Scale Questionnaire.

**Table 3 healthcare-13-02612-t003:** Correlation Analysis of Scale Scores in Cancer Caregivers.

	Zarit Caregiver Burden Sore	DASS-21- Depression	DASS-21- Anxiety	DASS-21- Stress
Zarit Caregiver Burden Sore	1	0.338 **	0.194 **	0.162 **
RSQ-Secure Attachment	−0.452 **	−0.017	−0.012	−0.057
RSQ-Dismissing Attachment	0.060	0.012	−0.064	0.016
RSQ-Preoccupied Attachment	0.276 **	0.019	0.055	−0.070
RSQ- Fearful Attachment	0.125 *	0.106 *	−0.055	0.053

* *p* < 0.05, ** *p* < 0.01; DASS-21, Depression Anxiety Stress Scale-21; RSQ, Relationship Scale Questionnaire Scale.

**Table 4 healthcare-13-02612-t004:** Correlation Analysis of Scale Scores in Caregivers of Individuals with Chronic Diseases.

	Zarit Caregiver Burden Sore	DASS-21- Depression	DASS-21- Anxiety	DASS-21- Stress
Zarit Caregiver Burden Sore	1	0.198 **	0.153 **	0.118 *
RSQ-Secure Attachment	−0.149 **	−0.136 *	−0.051	−0.109 *
RSQ-Dismissing Attachment	−0.036	0.021	−0.079	0.015
RSQ-Preoccupied Attachment	0.169 **	0.170 **	0.104 *	0.165 **
RSQ- Fearful Attachment	0.137 **	0.058	0.051	0.103 *

* *p* < 0.05, ** *p* < 0.01; DASS-21, Depression Anxiety Stress Scale-21; RSQ, Relationship Scale Questionnaire Scale.

**Table 5 healthcare-13-02612-t005:** Regression Analysis of Factors Associated With Caregiver Burden in Cancer Caregivers.

	Unstandardized			Standardized		
	B	SE	Lower	Upper	β	*p* Value
**Univariate regression analysis**						
**Age**	−0.043	0.022	−0.085	−0.001	−0.097	**0.049**
**DASS-21-Depression**	1.236	0.170	0.902	1.569	0.338	**<0.001**
**DASS-21-Anxiety**	0.517	0.129	0.264	0.771	0.194	**<0.001**
**DASS-21-Stress**	0.216	0.065	0.088	0.344	0.162	**0.001**
**RSQ-Secure Attachment**	−2.247	0.219	−2.677	−1.817	−0.452	**<0.001**
**RSQ-Dismissing Attachment**	0.430	0.356	−0.780	−0.270	1.129	0.228
**RSQ-Preoccupied Attachment**	1.141	0.196	0.756	1.527	0.276	**<0.001**
**RSQ-Fearful Attachment**	0.471	0.184	0.109	0.833	0.125	**0.011**
**Gender (Male vs. Female)**	1.586	0.456	0.690	2.482	0.169	**0.001**
**Type of Cancer (Non-common vs. common)**	1.800	0.453	0.910	2.690	0.193	**<0.001**
**Duration of Caregiving (>1 year vs. <1 year)**	1.195	0.451	0.309	2.081	0.130	**0.008**
**Relationship (Child vs. Others)**	1.972	0.485	1.018	2.926	0.197	**<0.001**
**Education (<College/PhD vs. ≥College/PhD)**	0.781	0.515	−0.231	1.793	0.075	0.130
**Multivariate regression analysis**						
**Age**	−0.006	0.024	−0.052	0.040	−0.013	0.805
**DASS-21- Depression**	0.982	0.139	0.708	1.256	0.269	**<0.001**
**DASS-21-Anxiety**	0.365	0.101	0.167	0.563	0.137	**<0.001**
**DASS-21-Stress**	0.170	0.051	0.070	0.269	0.127	**0.001**
**RSQ-Secure Attachment**	−1.909	0.190	−2.283	−1.535	−0.384	**<0.001**
**RSQ-Preoccupied Attachment**	0.769	0.159	0.457	1.081	0.186	**<0.001**
**RSQ-Fearful Attachment**	0.145	0.145	−0.139	0.430	0.039	0.316
**Gender (Male vs. Female)**	0.800	0.364	0.085	1.515	0.085	**0.028**
**Type of Cancer (non-common vs. common) ^1^**	0.324	0.113	0.101	0.547	0.112	**0.004**
**Duration of Caregiving (≥1 year vs. <1 year)**	0.948	0.368	0.225	1.670	0.103	**0.010**
**Relationship (Child vs. Others)**	1.379	0.531	0.335	2.424	0.138	**0.010**

^1^ “Common” refers to breast and lung cancers, while “non-common” encompasses all other types. DASS-21, Depression Anxiety Stress Scale-21; RSQ, Relationship Scale Questionnaire. Values in bold represent statistically significant differences (*p* < 0.05).

**Table 6 healthcare-13-02612-t006:** Regression Analysis of Factors Associated With Caregiver Burden in Chronic Disease Caregivers.

	Unstandardized			Standardized		
	B	SE	Lower	Upper	β	*p* Value
**Univariate regression analysis**						
**Age**	−0.086	0.040	−0.165	−0.008	−0.106	**0.032**
**DASS-21- Depression**	0.920	0.226	0.476	1.365	0.198	**<0.001**
**DASS-21-Anxiety**	0.528	0.170	0.806	0.194	0.861	**0.002**
**DASS-21-Stress**	0.250	0.105	0.044	0.456	0.118	**0.018**
**RSQ-Secure Attachment**	−1.132	0.373	−1.865	−0.399	−0.149	**0.003**
**RSQ-Dismissing Attachment**	−0.299	0.408	−1.101	0.503	−0.036	0.464
**RSQ-Preoccupied Attachment**	1.076	0.311	0.465	1.687	0.169	**0.001**
**RSQ-Fearful Attachment**	0.739	0.265	0.218	1.259	0.137	**0.006**
**Gender (Male vs. Female)**	2.603	0.742	1.145	4.061	0.172	**<0.001**
**Relationship (Child vs. Others)**	0.815	0.736	−0.631	2.261	0.055	0.268
**Education (<College/PhD vs. ≥College/PhD)**	−1.607	0.889	−3.353	0.140	−0.089	0.071
**Multivariate regression analysis**						
**Age**	−0.054	0.039	−0.131	0.022	−0.067	0.165
**DASS-21-Depression**	0.612	0.225	0.171	1.054	0.132	**0.007**
**DASS-21-Anxiety**	0.368	0.164	0.045	0.690	0.106	**0.026**
**DASS-21-Stress**	0.125	0.102	−0.076	0.326	0.059	0.223
**RSQ-Secure Attachment**	−0.138	0.553	−1.224	0.948	−0.018	0.803
**RSQ-Preoccupied Attachment**	0.663	0.385	−0.093	1.419	0.104	0.086
**RSQ-Fearful Attachment**	0.703	0.329	0.057	1.350	0.131	**0.033**
**Gender (Male vs. Female)**	2.781	0.737	1.332	4.229	0.184	**<0.001**

DASS-21, Depression Anxiety Stress Scale-21; RSQ, Relationship Scale Questionnaire. Values in bold represent statistically significant differences (*p* < 0.05)

## Data Availability

The data that support the findings of this study are not publicly available due to privacy concerns regarding participant data. However, the data are available from the corresponding author, I.O.U., upon reasonable request. The data are not publicly available due to privacy and ethical considerations concerning participant confidentiality.

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
