# Peer review of "Attachment Styles as Protective and Amplifying Factors in Caregiver Psychological Distress: A Multicenter Cross-Sectional Study in Oncology and Chronic Disease Caregiving"

_healthcare, 2025, doi:10.3390/healthcare13202612_

Round 1

Reviewer 1 Report

Comments and Suggestions for Authors

Dear Authors,

The article is valuable and relatively original in its investigation of the role of attachment styles in caregiver burden, a domain that is insufficiently explored compared to stress or burnout. The integration of attachment theory with caregiver burden analysis, together with the comparison between caregivers of palliative cancer patients and those of patients with chronic illnesses, provides a useful and nuanced comparative perspective. The use of validated instruments and moderation analyses adds rigor, and the conclusions may guide psychosocial interventions tailored to the caregiving context. Despite its limitations (cross-sectional design, culturally specific sample, and restrictive criteria), the study offers new data with practical relevance.

However, in the attached document I suggest several improvements and clarifications for your manuscript.

Reviewer 2 Report

Comments and Suggestions for Authors
  • Report the original psychometric validity of the instruments as well as the validity obtained through CFA with the target sample. For the latter, also report internal consistency using Cronbach’s alpha and McDonald’s omega, including confidence intervals.

  • Report the effect size in the results presented in Table 2.

  • For correlation analyses, interpret the strength of the associations in terms of weak, moderate, strong, or very strong (see https://doi.org/10.1017/CBO9780511761676).

  • It is recommended to complement the regression analyses by reporting the Variance Inflation Factor (VIF), serial correlation using the Durbin-Watson test, and an analysis of influential values.

Reviewer 3 Report

Comments and Suggestions for Authors

Thank you very much for inviting me to review the manuscript titled “Attachment Styles as Protective and Amplifying Factors in Caregiver Psychological Distress: Multicenter Insights from Oncology and Chronic Disease Caregiving.” The article is interesting, attempts to answer an important research question, and I appreciate the authors ' efforts. However, numerous concerns that require revisions.

  1. The title reflects the contents and exposure assessment of the manuscript. However, please review the STROBE checklist.
  2. Abstract: Well-structured. Please check the word count according to the Healthcare, MDPI. I suggest including key statistical results with confidence intervals rather than p-values alone to emphasize the precision of estimates
  3. The introduction section is well-written, provides strong background information, and a rationale is given. Literatures are cited properly. However, I feel that it could be slightly more focused (some sections are overly verbose and repetitive).
  4. I appreciate the authors for their efforts to conduct a multicenter study using a standard methodology. However, what is the status of potentially important confounders (e.g., caregivers’ own health status, income, social support, hours spent caregiving)?
  5. Another important aspect is missing data exclusion. Did the author do any missing data analysis (not related to the present study results)? Because it would have significant selection bias.
  6. Also, the study may lack external validity.
  7. Please check the STROBE statement and reorganise in presenting the methods.
  8. Also, I see that the ethics statement is in the manuscript (after the references section). Nonetheless, it is always preferable to mention the ethics statement, recruitment strategy, etc, as a part of the methods.
  9. Results are generally presented well. But, p-values without reporting effect sizes with CIs reduce interpretability.
  10. Discussions are fine and linking findings to existing literature. However, several sections are overly descriptive/narrative. Please make a more critical analysis and mention the implications for public health.
  11. Overall, the English quality is fine. Please incorporate these comments.

Round 2

Reviewer 1 Report

Comments and Suggestions for Authors

Dear Authors,

Thank you for your responses to my suggestions for improvement. I have carefully reviewed them and found that you have adequately addressed all the requested revisions, and the changes made contribute significantly to improving the quality of the manuscript. I have no further comments.

Author Response

Dear Reviewer,

We would like to express our sincere gratitude for your time and effort in reviewing our manuscript.

We greatly appreciate your thoughtful comments and constructive feedback during the review process. We are pleased to hear that our revisions have adequately addressed your suggestions and that the changes have improved the quality of the manuscript.

Your guidance has been invaluable in enhancing the clarity and rigor of our work. Thank you once again for your support and for helping us strengthen the manuscript.

With best regards

Reviewer 3 Report

Comments and Suggestions for Authors

Dear authors,

Thanks for making tremendous efforts for revising the manuscript. Highly appreciated. The manuscript's quality has improved significantly. 

However, I have a couple of minor comments. Firstly, the introduction can be improved/trimmed further. Attaching the STROBE checklist would enhance transparency.

Wish you all the best. 

Author Response

Dear Reviewer,

We sincerely thank you for your continued time and thoughtful feedback on our manuscript.

We greatly appreciate your kind words regarding the improvements made to the manuscript and are grateful for your encouragement and guidance throughout the review process.

In response to your final suggestions:

We have carefully reviewed the Introduction and revised it to enhance clarity and brevity. Redundant content has been removed, and the narrative flow has been improved to better focus on the study’s rationale and objectives.

As recommended, we have completed the STROBE checklist for cross-sectional studies and included it as a supplementary file to improve reporting transparency.

We hope these final revisions address your suggestions fully. Thank you once again for your valuable input and support in strengthening the manuscript.

With kind regards